# Implementing communication and decision-making interventions directed at goals of care: a theory-led scoping review

Amanda Cummings,[1,2] Susi Lund,[1,2] Natasha Campling,[1,2] Carl May,[1,2,3] Alison Richardson,[1,2,3] Michelle Myall[1,2]

## ABSTRACT

**Objectives** To identify the factors that promote and inhibit the implementation of interventions that improve communication and decision-making directed at goals of care in the event of acute clinical deterioration.

**Design and methods** A scoping review was undertaken based on the methodological framework of Arksey and O'Malley for conducting this type of review. Searches were carried out in Medline and Cumulative Index to Nursing and Allied Health Literature (CINAHL) to identify peer-reviewed papers and in Google to identify grey literature. Searches were limited to those published in the English language from 2000 onwards. Inclusion and exclusion criteria were applied, and only papers that had a specific focus on implementation in practice were selected. Data extracted were treated as qualitative and subjected to directed content analysis. A theory-informed coding framework using Normalisation Process Theory (NPT) was applied to characterise and explain implementation processes.

**Results** Searches identified 2619 citations, 43 of which met the inclusion criteria. Analysis generated six themes fundamental to successful implementation of goals of care interventions: (1) input into development; (2) key clinical proponents; (3) training and education; (4) intervention workability and functionality; (5) setting and context; and (6) perceived value and appraisal.

**Conclusions** A broad and diverse literature focusing on implementation of goals of care interventions was identified. Our review recognised these interventions as both complex and contentious in nature, making their incorporation into routine clinical practice dependent on a number of factors. Implementing such interventions presents challenges at individual, organisational and systems levels, which make them difficult to introduce and embed. We have identified a series of factors that influence successful implementation and our analysis has distilled key learning points, conceptualised as a set of propositions, we consider relevant to implementing other complex and contentious interventions.

[1]Faculty of Health Sciences, University of Southampton, Southampton, UK
[2]NIHR CLAHRC Wessex, University of Southampton, Southampton, UK
[3]University Hospital Southampton, NHS Foundation Trust, Southampton, UK

**Correspondence to**
Dr Amanda Cummings;
ac2a13@soton.ac.uk

## Strengths and limitations of this study

► This paper outlines a scoping review of a broad and diverse literature, both published and grey, focusing on the implementation of goals of care interventions for patients facing clinical deterioration. However, because of its focus on implementation, not all examples of goals of care interventions could be included.

► Normalisation Process Theory (NPT) was used to investigate and explain the successful implementation of interventions, and a theoretical approach has been applied to all stages of the review process.

► In a number of included papers, information relating to implementation barriers was missing, and there was bias towards the presentation of positive outcomes. This may reflect a reluctance to focus on challenges with study authors keen to exhibit and promote the benefits of interventions.

► The review led to goals of care interventions being defined as 'contentious' with a moral purpose and value, and identified the elements and learnings that could be transferable to other examples of such interventions.

► We have proposed that contentious interventions consist of components at three levels. Across the literature reviewed, the focus was on components at the individual level where negotiated decision-making between participants occurs. There was limited focus on the components that take place within and across organisations and the influence of system constraints. This has important limitations for our interpretation of data and analysis.

## INTRODUCTION

In the event of a patient becoming acutely unwell, treatment and care decisions are recommended by clinicians. While these decisions are based on the clinical judgement of a healthcare professional, they should also be bound by the preferences and wishes of the patient and their family. Processes and tools (referred to here as goals of care interventions), that provide a framework for discussing and documenting appropriate treatment options in the event of acute clinical deterioration are paramount.

These interventions aim to improve patient and family involvement, enabling exploration and understanding of the current clinical situation and facilitation of communication and negotiated decision-making about future treatment options.[1 2] They offer a means for patients' preferences to be taken into account, improving communication and clarity across the wider clinical team.

We refer to goals of care in the event of acute deterioration, where different levels of treatment might be appropriate and range from full escalation in a critical care environment to symptom control measures.[3] Goals of care are currently referred to using various terms, including but not confined to ceilings of care, treatment escalation plans and treatment limitations. They exist in numerous formations including a specific, dedicated paper form, a narrative entry in a paper medical record and inclusion in an electronic patient record, and may be introduced as an extension of the 'do not attempt cardio-pulmonary resuscitation' (DNACPR) process. These interventions offer a system for recording recommended treatment and care, ideally including all components of the decision pathway and can be applied across different care settings. They require a process to be created and implemented that is recognised across organisations, takes account of sociolegal frameworks, such as the UK Mental Capacity Act (MCA) (2005),[4] and is designed to protect individuals who do not have capacity to make decisions about their care and treatment.

Goals of care interventions are complex and consist of multiple interacting components. The number and difficulty of behaviours required by those delivering or receiving such an intervention, variability of outcomes, and degree of flexibility and tailoring that is permitted, all contribute to making this a complex intervention.[5] As well as being complex, goals of care interventions are established as interactions and recording systems with a moral purpose and value. This means they may be *contentious* in practice because they contain elements that seek to routinise highly complex clinical skills, practice and different types of wisdom, in a context of uncertainty. Contentiousness can arise because the intervention relies on patient, family and clinician interactions, interclinician interactions (potentially across clinical settings and organisational boundaries), and societal and legal frameworks, such as the MCA (2005) and the European Convention on Human Rights (2002).[4 6]

In spite of a growing body of literature describing the introduction and benefits of goals of care interventions, little is known about the factors that influence their successful implementation in clinical practice. Here implementation is defined as 'any deliberately initiated attempt to introduce new, or modify existing, patterns of action in health care or some other formal organizational setting. Deliberate initiation means that an intervention is: institutionally sanctioned; formally defined; consciously planned; and intended to lead to a changed outcome'.[7] As we have previously argued, "this is more than the adoption or diffusion of innovations"

as effective implementation is about interventions being made workable and embedded in routine clinical practice.[7]

Understanding and evaluating the implementation of complex interventions in practice remain a challenge for healthcare managers, policy makers and for those who enact them outside of formal research settings.[8] Furthermore, as interventions found to be effective in the context of health services research studies can fail to translate into meaningful healthcare outcomes across varying contexts,[9] this makes understanding the reasons for failure or partial success even more essential. Implementation science, which promotes the integration of research findings and evidence into healthcare policy and practice, is increasingly recognised within health services research to make a key contribution to such knowledge. Comprehensive process evaluation of the implementation of healthcare interventions is increasingly important for future learning,[10] as it enables understanding of transition from closed systems of highly structured research or service development projects into the real world of open systems healthcare delivery where they are operationalised.[11 12] Learning from the available existing knowledge in this area can be used to inform healthcare practice change and contribute to the field of implementation science.

Having previously applied Normalisation Process Theory (NPT) to aid learning in the comparable context of advance care plans,[13] we have used NPT to characterise and explain implementation processes.[14] NPT provides a set of tools to investigate and understand the processes through which interventions are operationalised in healthcare settings and incorporated into everyday practice.[14–17]

In this paper, we present a scoping review of goals of care interventions which aims to *identify the factors that promote and inhibit the implementation of interventions which improve communication and decision-making directed at goals of care.* Using goals of care interventions as an example, a secondary aim of this review is to *characterise the components and consider the implications for implementation of, contentious interventions.*

## DESIGN AND METHODS
A scoping review was the most appropriate methodology, given the need to extract and map principles from a diverse and broad body of evidence.[18 19]

We used Arksey and O'Malley's five-stage framework for conducting scoping reviews, which includes identifying the research question, identifying relevant literature, selection, charting the data and collating, summarising and reporting the results.[20] This guided the scoping review and where necessary we developed more specific procedures to inform the review process. Levac *et al*'s[21] recommendations for refining the methodological application were also incorporated to increase rigour of the review process.

**Table 1** Inclusion and exclusion eligibility criteria

| Inclusion criteria | Exclusion criteria |
|---|---|
| Literature describing implementation of interventions related to communication and decision-making around goals of care. | Papers not describing an intervention (process or tool) |
| Studies with adult patients in hospital and community settings | Papers reporting treatment effectiveness |
| Studies involving end-of-life care, clinical deterioration and clinically uncertain outcomes | Papers describing do not attempt cardiopulmonary resuscitation orders only |
| Studies published in the English language | Studies in neonatal and paediatric settings |
| Papers published between 2000 and 2015 | Studies involving brain stem death |
| Qualitative and quantitative studies, including clinical trials and randomised controlled trials | Studies using biomedical data and drug trials |
| Published conference abstracts/conference-related papers | Non-English-language studies |
| Grey literature (limited to policies, reports, research posters, patient/staff guidance, websites) | Papers published before 1 January 2000 |

## Inclusion and exclusion criteria

Literature was selected using specific inclusion and exclusion criteria (table 1), and was included if it was considered to have a specific focus on implementation in practice. To identify all relevant literature on goals of care, grey literature was included and actively sought as part of the search strategy. Only papers published from the year 2000 onwards were included as those published earlier were unlikely to reflect current practice. Papers focusing solely on DNACPR orders were excluded due to their focus on only one decisional element of goals of care. Existing evidence also suggests that DNACPR decisions are not always discussed with patients or families.[22] The combining of DNACPR decisions within wider goals of care interventions has been shown to improve clarity and communication, and the focus of this review is on implementing a process whereby goals of care are discussed.[23]

## Search strategy and information sources

The search strategy was designed to identify primary studies and other literature, both published and unpublished, that met the eligibility criteria. Separate searches were undertaken for primary and grey literature.

### Primary literature searches

The primary literature search was carried out in two stages. The first stage involved an initial search of the bibliographical database Medline using a preliminary keyword search based on the terms of the topic, the text terms used in titles and abstracts, and index terms used to describe articles. Full details of the primary literature search strategy are outlined in online supplementary appendix 1.

Terms identified in the titles and abstracts of relevant articles produced from stage 1 were used to develop further keywords for the second stage of the literature search (see online supplementary appendix 1). Dual combined keyword searches were conducted in Medline and Cumulative Index to Nursing and Allied Health Literature (CINAHL) bibliographical databases, with a third keyword added if searches produced >200 results. In addition, further literature was identified through existing knowledge and networks.

### Grey literature searches

Google searches were conducted using terminology relating to known goals of care interventions identified in an earlier scoping exercise which mapped the use of forms for recording these decisions in the UK. The first 10 results were screened for relevance and further review. A further Google search focused on policies and guidance related to goals of care was undertaken and the first 50 results screened for relevance.

All screened, de-duplicated citations were imported into the bibliographical software management package EndNote. Searches were completed by August 2015. This was due to the imminence of a national programme of work which has led to the Recommended Summary Plan for Emergency Care and Treatment (RESPECT).[24] Communication and decision-making around goals of care is a growing area of interest, and it is hoped that findings from this review can be used to inform the implementation of such a major advancement in the field.

### Screening

An extensive screening process was undertaken. At the first stage of screening, articles were assessed by two independent reviewers (AC, CRM) based on the information provided in the title. Primary literature judged to be relevant after first screen, and which met the eligibility criteria, were obtained in full text. Articles for which there was disagreement between reviewers were also obtained in full text. Full-text articles were examined for adherence to the inclusion criteria and then screened (by AC, with input from CRM and MM, in cases of uncertainty or disagreement). Grey literature were title-screened for relevance and a further full review examined adherence to inclusion criteria.

## Quality assessment

This scoping review included a non-heterogeneous sample of primary and grey literature, which made it difficult to universally apply quality assessment criteria. As standard to most scoping reviews, we did not undertake formal quality assessment and excluded papers only on grounds of relevance. As a result the analytical focus of this review centred on a critique of relevance and contribution of the included literature and did not consider methodological quality.

## Data extraction

In line with Arksey and O'Malley's method,[20] data extraction (charting) was multistaged. In the first stage we collected descriptive characteristics from each paper, such as study design and setting. In the second stage, findings and discussion sections of included literature were extracted into a data extraction tool (see online supplementary appendix 2) informed by NPT.[15–17]

The data extraction tool was designed to chart specific details of the literature and to understand factors influencing implementation. The tool was piloted on a sample (n=5) of primary and grey literature, with subsequent amendments resulting in the final version. Data were extracted by independent reviewers (AC, SL, MB). Following Levac et al's[21] recommendations, two reviewers (AC, MM) independently extracted data for a 30% sample of primary literature to ensure approaches to extraction were consistent with each other and with the research aims.

## Data analysis

A two-stage analytical process was undertaken. During the first stage, data extracted were treated as qualitative data and analysed using directed content analysis.[25] A theory-informed coding framework was developed using the four main constructs and subconstructs of NPT.[15 16] Data were identified and categorised to the constructs and subconstructs of NPT, exploring barriers and facilitators to implementation.

The use of NPT as a theoretical framework followed its successful application in a number of different healthcare intervention reviews.[13 26–30] NPT investigates and explains the successful operationalisation of interventions: how they become part of everyday practice in healthcare settings. It embodies the different types of 'work' undertaken by individuals around implementing, embedding and integrating, and allows us to understand the social structures and contexts through which new interventions are operationalised.[14–17] In relation to implementation of goals of care interventions, definitions of the four core constructs and subconstructs of NPT used in this review are outlined in online supplementary appendix 3.

During the second stage, thematic analysis of the literature content was undertaken to identify, characterise and explain the factors that shape implementation of interventions that initiate communication and decision-making around goals of care. A final higher level of analytical interpretation followed, to characterise the components of contentious interventions and generate transferable learning outcomes for their implementation.

## RESULTS

Searches identified 2619 citations. Following de-duplication and relevance screening, 43 sources of literature (relating to 23 interventions), including 24 items of published literature (eg, peer-reviewed papers, conference abstracts) and 19 items of grey literature (eg, conference posters, patient information documents), met the inclusion criteria and were included for data extraction (see figure 1). Table 2 provides a summary of the characteristics of included literature, the range of interventions described and the decisions of interest addressed, including DNACPR, goals of active care, supportive or palliative care, and those that are limited to communication guidelines only.

Thematic analysis of literature content suggested there were six common themes fundamental to the successful implementation of goals of care interventions: input into development, key clinical proponents, training and education, intervention workability and functionality, setting and context, and perceived value and appraisal. These are outlined below. The links between the subconstructs of NPT and these identified themes are outlined in table 3. A summary of results by NPT construct is presented in online supplementary appendix 4 as a summary and appraisal of the literature.

### Input into development

The involvement of clinical staff in the development of interventions facilitated the identification of current shortfalls in practice, understanding purpose, and shaping novelty, accessibility and utility of design.[31–35] Developmental input also assisted in promoting' buy-in', legitimisation and confidence in the intervention.[31–34 36–41] While some literature focused on the importance of input from senior, specialist individuals,[32 33 36 41] including those within implementation settings,[33 34 42] for intervention design, multidisciplinary collaboration in the development process was common across papers[3 31 33 34 36 39–45] and helpful for gaining collective knowledge from all potential users.[31 34 36 39 40] In one paper, individuals who had expressed disagreement with one intervention were actively sought for involvement at each stage of development,[31] highlighting how inclusion in development can contribute to overcoming barriers of individual resistance. Reconfiguration of forms and procedures was incorporated in the development and implementation process in a number of initiatives.[31 33 34 36 42] The benefit of this approach was that it allowed for assessment of factors such as usability, clarity and safety, incorporating feedback into further iterations.[31 33 42]

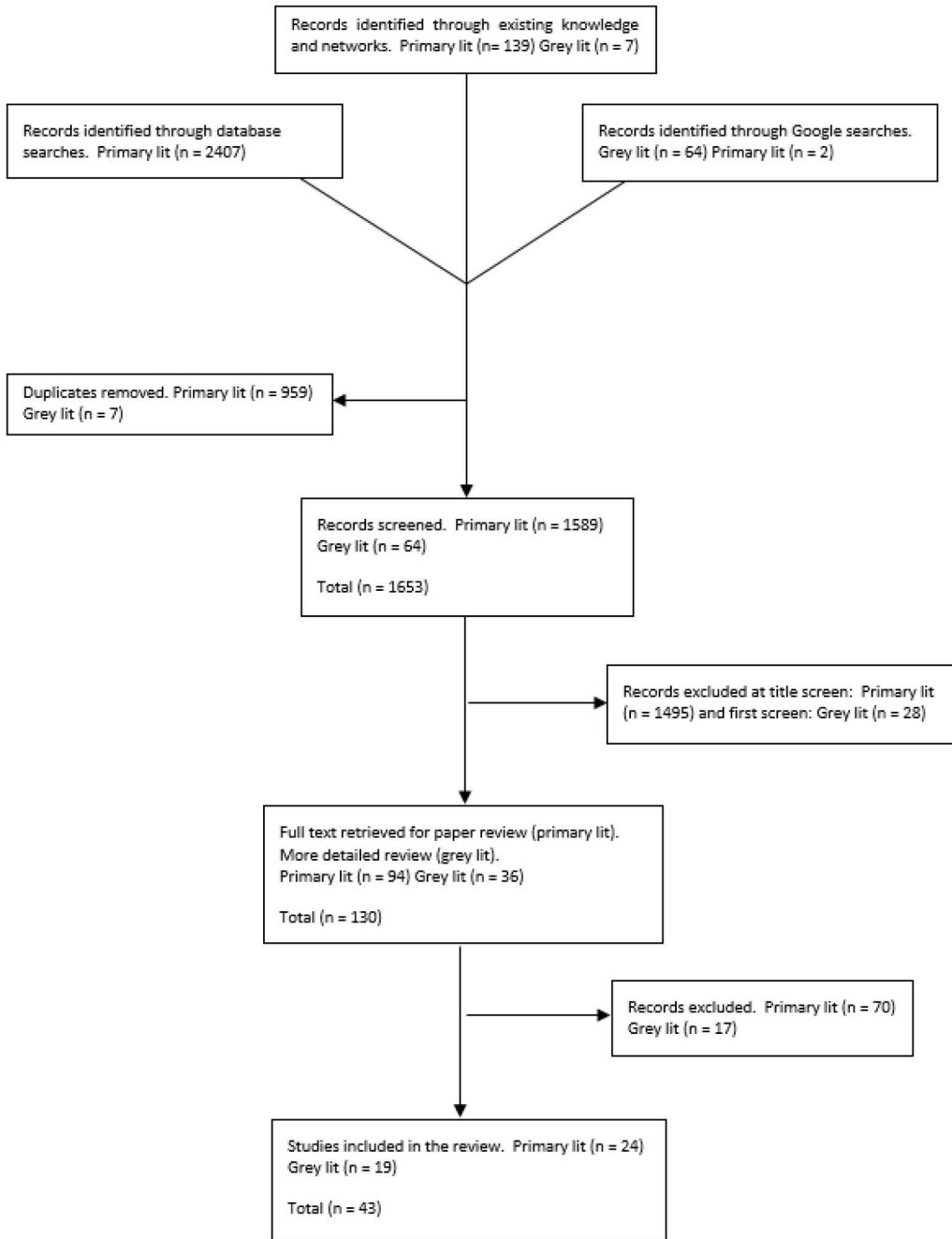

**Figure 1** Flow diagram of literature screening.

### Key clinical proponents

Key clinical proponents were assigned to orchestrate and lead implementation within specific sites or settings.[32 42 46 47] They were key resources for attending and cascading training, as well as overseeing changes, embedding cultural change and providing mentorship.[32 42 46–48] Their direct contact with intervention instigators appeared important in orientating and maintaining momentum of local leadership, including training and facilitation.[32 42 47] Similarly, support from individual senior managers in ensuring compliance with training and interventions, promoting sustained use, as well as acting as catalysts for initial implementation was valuable.[32 35 42 46 47 49] Trust level managerial support was also evident through incorporation into policy and protocol[34 45 49–52] and approval from directors.[39 40] Managerial backing could be influential for organisational change, although in reality, engagement with the management of healthcare organisations was recognised by some for its complexity.[32 42]

### Training and education

Training and education encouraged 'buy-in',[3 31 32 36 39 42 46 47 51 52] promoted understanding of tasks and responsibilities,[31 36 39 42 47 51] and facilitated a shared understanding of purpose.[44 47 51] Staff often negatively

**Table 2** Summary of included literature

| Name and description of intervention | Decisions of interest | | | | Author and year | Country and healthcare setting | Literature type | Study design and methods | Study objective |
|---|---|---|---|---|---|---|---|---|---|
| | DNACPR | Ceilings of active care | Supportive/Palliative care | Communication guidelines around goals of care | | | | | |
| *AMBER care bundle* Care bundle consisting of a proforma and sticker. Promotes consistent communication and care planning for patients identified as facing uncertain recovery | X | X | | | Carey et al, 2014[36] | UK Hospital (five acute wards) | Published peer-reviewed study | Descriptive/Quantitative: Overview of development and implementation of AMBER, and data on clinical impact | To describe the design, development and implementation of the AMBER care bundle and its impact on clinical service |
| | | | | | Etkind et al, 2014[60] | UK Hospital (five acute wards) | Published peer-reviewed study | Quantitative: Retrospective electronic case note review of patients who died in wards where AMBER was implemented | To evaluate application of the AMBER care bundle in a UK hospital and describe factors affecting its use |
| | | | | | Morris et al, 2011[48] | UK Hospital | Published conference abstract | Descriptive: Overview of AMBER and its implementation | Learning points from implementation of the AMBER care bundle |
| | | | | | Doncaster and Bassetlaw Hospitals NHS Foundation Trust, 2015[50] | UK NHS Trust | Grey literature | Policy document | Management of patients who are at end of life |
| | | | | | Guy's and St Thomas' NHS Foundation Trust, 2012[55] | UK NHS Trust | Grey literature | Patient guidance | The AMBER care bundle, a guide for patients, their relatives and carers |
| *Universal Form of Treatment Options (UFTO)* Form contextualising the CPR decision within overall treatment plans. Includes documentation and discussion of care and treatment decisions | X | X | X | | Fritz et al, 2013[3] | UK Hospital (two acute wards) | Published peer-reviewed study | Prospective mixed method: Case note review of DNACPR patients, qualitative interviews and observations | To determine whether the introduction of UFTO reduces harms in patients in whom a DNACPR was made, and to understand the mechanisms for observed change |
| | | | | | Fritz et al, 2015[31] | UK Hospital (two acute wards) | Published peer-reviewed study | Expert consensus: Cross-disciplinary adapted Delphi method | To describe a cross-disciplinary approach to developing a universal form of treatment options as an alternative to DNACPR orders |
| | | | | | www.ufto.org Accessed: 1 July 2015[78] | UK | Grey literature | Website | UFTO: universal form of treatment options |
| | | | | | www.ufto.org/background/scientific-background Accessed 1 July 2015[43] | UK | Grey literature | Website | UFTO: scientific background |
| *Goals of Patient Care Summary (GOPC)* Form to summarise goals of care. Serves as a guide for treatment decisions as the patient's condition improves, stabilises or deteriorates | X | X | X | | Brimblecombe et al, 2014[53] | Australia Hospital (adult medical inpatients) | Published peer-reviewed study | Quantitative: Cross-sectional review of medical inpatients and audit of inpatients requiring emergency medical review | To review the implementation of the GOPC: assess GOPC uptake, quantify completion and review outcomes of the GOPC assessment |

Continued

**Table 2** Continued

| Name and description of intervention | Decisions of interest | | | | Author and year | Country and healthcare setting | Literature type | Study design and methods | Study objective |
| --- | --- | --- | --- | --- | --- | --- | --- | --- | --- |
| | DNACPR | Ceilings of active care | Supportive/Palliative care | Communication guidelines around goals of care | | | | | |
| *Physician Orders for Life-Sustaining Treatment (POLST)* Form to ensure that patient wishes regarding life-sustaining treatments are honoured. Converts treatment preferences into medical orders | X | X | X | | Hickman et al, 2009[54] | USA Hospice (hospices in 3 US states) | Published peer-reviewed study | Descriptive/Quantitative: Survey of hospice staff members; case note review of deceased patients at subsample of POLST-using hospices | To evaluate use of POLST by hospice programmes, attitudes of hospice staff towards POLST, the effect of POLST on the use of life-sustaining treatments and the treatment options selected by patients |
| *Gold Standards Framework for Care Homes (GSFCH) and an adapted Liverpool Care Pathway (LCP) for Care Homes* The implementation of two end-of-life care tools: the GSFCH supportive/palliative care register and an adapted LCP for Care Homes | X | X | X | | Hockley et al, 2010[32] | UK Nursing home (seven nursing homes in Scotland) | Published peer-reviewed study | Quantitative/In-depth evaluation: Overview of implementation, retrospective case note review of deceased residents and staff audit | To evaluate the effectiveness of a high facilitation implementation on professional practices and residents outcomes |
| *UIHC (University of Iowa Hospital and Clinics) Policy Regarding Do Not Resuscitate Orders and End of Life Discussions with Hospitalised Patients* DNACPR order policy placing CPR discussions within a framework of end-of-life discussions, and within a larger goal-orientated context | X | X | X | | Kaldjian and Broderick, 2011[51] | USA Hospital (academic medical centre) | Published peer-reviewed study | Descriptive: Overview of the integration of goals of care into staff education and revision of hospital DNACPR policy; literature review and survey of hospitalised adults | To describe the background and rationale for DNACPR policy revision and development; to describe the policy itself and ongoing implementation efforts |
| *A revised DNACPR form, decision making tools and updated CPR guidelines* | X | X | X | | Mbriwa and Limaye, 2014[59] | UK Hospital (elderly and acute medical inpatients) | Published conference abstract | Quantitative: Prospective case note audit | To assess improvement in CPR decision-making and ceiling of care documentation/discussions since the introduction of a revised DNACPR form, decision-making tools and guidelines |

Continued

**Table 2** Continued

| | Decisions of interest | | | | | | | | |
| Name and description of intervention | DNACPR | Ceilings of active care | Supportive/Palliative care | Communication guidelines around goals of care | Author and year | Country and healthcare setting | Literature type | Study design and methods | Study objective |
|---|---|---|---|---|---|---|---|---|---|
| *ProvidersSignout for Scope ofTreatment (PSOST)* A daily handover tool documenting resuscitation status, goals of care, desired and undesired interventions, in addition to standard handover information | X | X | X | | Newport et al, 2010[37] | USA Palliative care (inpatient unit) | Published peer-reviewed study | Descriptive: Overview of tool development, implementation and utility | To describe the PSOST tool and evaluate its implementation and use in a palliative care unit |
| *An intervention informing clinicians of the laws on Patients' Rights and the importance of advanced care planning, alongside a new DNR (do not resuscitate) form* Adapted DNACPR form, putting emphasis on the motivation, participants involved and communication of the decision-making process | X | X | | | Piers et al, 2014[38] | Belgium Hospital (adult inpatients) | Published peer-reviewed study | Quantitative: Prospective observational study of deceased inpatients preintervention and postintervention; cross-sectional study of patients admitted to non-intensive wards | To compare the quality of end-of-life care and DNACPR decisions between hospital wards and after a hospital-wide intervention to improve advanced care planning |
| *Treatment Escalation Plan and Resuscitation Decision (TEP-CPR)* Treatment escalation plan tool, to be included in the resuscitation form | X | X | X | | Stockdale et al, 2013[33] | UK Hospital (two acute wards) | Published peer-reviewed study | Quantitative/Descriptive: Case note audit and staff survey, followed by overview of TEP-CPR development; staff survey of out of hours treatment in patients with TEP-CPR and controls | To develop and test a tool for escalation planning; to identify whether a TEP-CPR could reduce the number of patients receiving inappropriate treatments out of hours |
| *Ceiling ofTreatment (COT) Form* Tool for improving decision-making and documentation of treatment escalation decisions | X | X | X | | Dahill et al, 2013[34] | UK Hospital (elderly care wards) | Published peer-reviewed study | Quantitative/Descriptive: Overview of COT development and implementation; preimplementation and postimplementation case note review of treatment escalation decisions and staff feedback survey | To develop, pilot and implement a tool to improve decision-making and the documentation of treatment escalation decisions |
| | | | | | Powter et al, 2012[39] | UK Hospital (elderly care wards) | Grey literature Poster | Quantitative/Descriptive: Overview of COT development and implementation; prepilot and postpilot data on decision documentation and staff survey | To improve decision-making and documentation of CPR and ceiling of treatment decisions |

Continued

**Table 2** Continued

| Name and description of intervention | DNACPR | Ceilings of active care | Supportive/Palliative care | Communication guidelines around goals of care | Author and year | Country and healthcare setting | Literature type | Study design and methods | Study objective |
|---|---|---|---|---|---|---|---|---|---|
| | | Decisions of interest | | | | | | | |
| *Treatment Escalation Plan (TEP) (Devon)* Form outlining what treatment options would be appropriate if a patient were to become acutely unwell | X | X | | | Obolensky et al, 2010[1] | UK Hospital (two acute trauma & orthopaedic wards) | Published peer-reviewed study | Prospective quantitative and qualitative evaluation: Patient/relative interviews to evaluate the TEP process | To evaluate patient and relative experiences of, and thoughts regarding the Devon TEP |
| | | | | | Mercer, 2009[61] | UK Hospital | Published letter | Letter to the editor Descriptive: Overview of TEP process | The death of DNR: TEPs |
| | | | | | Obolensky and Mercer, 2007[66] | UK Hospital | Published conference abstract | Descriptive/Quantitative: overview of TEP process and staff survey | To evaluate initial impact of the Devon TEP from a clinical staff perspective |
| | | | | | Obolensky and Mercer, 2007[56] | UK Hospital | Published conference abstract | Descriptive/Quantitative: overview of TEP process and patient survey | To evaluate patient experiences of the Devon TEP |
| | | | | | Karakusevic[44] | UK Hospital (acute hospital) | Grey literature | Poster Descriptive: Overview of background to development and TEP implementation | To evaluate implementation of the Devon TEP |
| | | | | | Torbay and Southern Devon Health and Care NHS Trust, 2014[52] | UK NHS Trust | Grey literature | Policy document | TEP and resuscitation decision records policy |
| | | | | | Care Quality Commission (CQC), 2014[46] | UK NHS Trust | Grey literature | CQC report | Northern Devon Healthcare NHS Trust, end-of-life care quality report |
| | | | | | Rowcroft Hospice, 2013[57] | UK Hospice | Grey literature | Patient guidance | TEP forms, public information document |
| | | | | | Devon Local Medical Committee, 2013[79] | UK | Grey literature | Staff guidance | TEP—newsletter update article |
| | | | | | Macmillan Cancer Support, St Luke's Hospice Plymouth & North, East and West Devon Clinical Commissioning Group[80] | UK Hospice | Grey literature | Staff guidance | New TEP form (V.10)—frequently asked questions document |
| | | | | | North Devon Healthcare NHS Trust[81] | UK NHS Trust | Grey literature | Staff guidance | Staff briefing note on the use of TEPs |
| | | | | | NHS Improving Quality Team, 2014[82] | UK | Grey literature | Staff guidance | Guidance for completing TEP and resuscitation decisions |
| *Treatment Escalation Plan (TEP)* Tool for planning the care of a patient at risk of deterioration | X | X | X | | Paes and O'Neill, 2012[62] | UK Hospital (six acute specialties) | Published conference abstract | Descriptive/Quantitative: Pilot, audit and staff feedback survey | To evaluate staff feedback and decision-making outcomes before and after TEP introduction |
| | | | | | Paes, 2012[63] | UK Hospital (six acute wards and community hospital) | Grey literature | Poster Descriptive/Quantitative: Overview of TEP implementation and next steps: case note review audit before and after TEP pilot, and staff survey | To pilot TEP and evaluate staff feedback, out of hours decision-making/discussions and TEP completion outcomes; to determine whether and how TEP should be introduced across the trust |

Continued

**Table 2** Continued

| Name and description of intervention | DNACPR | Ceilings of active care | Supportive/Palliative care | Communication guidelines around goals of care | Author and year | Country and healthcare setting | Literature type | Study design and methods | Study objective |
|---|---|---|---|---|---|---|---|---|---|
| | | | | **Decisions of interest** | | | | | |
| HDU (high dependency unit) Specific Treatment Escalation Plan Plan for ceiling of treatment and escalation level decision on every patient within 24 hours | X | X | | | Hannah, 2014[40] | UK Hospital (medical HDU) | Grey literature | Poster Descriptive: Overview of TEP development, implementation and next steps, including informal staff feedback | To describe and evaluate the process of TEP development and its implementation in a medical high dependency unit |
| Treatment Escalation Plan (TEP) Proforma | X | X | X | | Thomson et al, 2014[64] | UK Hospital (acute medicine) | Grey literature | Poster Quantitative: Case note review audit before and after TEP pilot, and staff survey | To pilot TEP and evaluate staff feedback and DNACPR/TEP documentation outcomes |
| Cornwall Treatment Escalation Plan and Resuscitation Decision Record | X | X | | | Royal Cornwall Hospitals NHS Trust, 2015[49] | UK NHS Trust | Grey literature | Policy document | TEP and resuscitation decision record policy |
| Escalation plan and resuscitation status (EPARS) Form amalgamating escalation plan and CPR decisions | X | X | X | | West Suffolk NHS Foundation Trust, 2013[45] | UK NHS Trust | Grey literature | Policy document | Trust policy and procedure: EPARS (including DNACPR) |
| Treatment Escalation Plan (TEP) | X | X | | | Basildon and Thurrock University Hospitals NHS Foundation Trust, 2014[58] | UK NHS Trust | Grey literature | Patient guidance | TEP patient information |
| Content guidelines for discussions of CPR and life-sustaining therapy Guidelines for the content of a discussion about resuscitation or goals of care | X | X | | X | Downar and Hawryluck, 2010[41] | Canada Hospital (12 physicians) | Published peer-reviewed study | Expert Consensus: Delphi method to develop content guidelines | To develop content guidelines for discussions of CPR and life-sustaining therapy |
| A palliative care treatment plan intervention Best practice guidance supported by a staff education programme, electronic decision support tool and paper educational tools | X | | X | | Bailey et al, 2014[47] | USA Hospital (inpatient units at six veteran medical centres) | Published peer-reviewed study | Descriptive/Quantitative: Overview of intervention and its multicentre implementation; preimplementation and postimplementation case note review of end of life care in deceased patients | To evaluate the effectiveness of a multimodal intervention strategy to improve the processes and quality of end-of-life care in acute inpatient settings |

Continued

**Table 2** Continued

| | Decisions of interest | | | | | | | | |
| Name and description of intervention | DNACPR | Ceilings of active care | Supportive/Palliative care | Communication guidelines around goals of care | Author and year | Country and healthcare setting | Literature type | Study design and methods | Study objective |
|---|---|---|---|---|---|---|---|---|---|
| *The Palliative Care for Advanced Disease (PCAD) pathway* To improve palliative care for inpatients who are expected to die from advanced disease; clinical pathway including: –interdisciplinary care pathway –nurses documentation flowsheet –physicians order sheet with guidelines for symptom management | | | X | | Bookbinder *et al*, 2005[42] | USA Hospital (3 inpatient units) | Published peer-reviewed study | Descriptive/Quantitative: Overview of intervention, its development and implementation; case note review of care outcomes in patients who died before and during implementation, in both intervention and control units | To describe the development and quality improvement strategy for the implementation of PCAD; to pilot and evaluate the utility of PCAD |
| *Serious illness conversation guide* Guide describing key elements to be addressed in patient conversations about serious illness care goals | | | X | X | Bernacki and Block, 2014[65] | USA | Published peer-reviewed study | Literature review and synthesis of best practices in conversations about serious illness care goals | To provide clinicians with practical, evidence-based advice (in the form of a serious illness conversation guide) |
| *The VOICE (Values Options in Cancer Care) intervention* Education and coaching communication intervention To facilitate communication and decision-making among oncologists, patients and caregivers | | | | X | Hoerger *et al*, 2013[35] | USA Oncology services | Published peer-reviewed study | Descriptive: Overview of intervention and its planned implementation as part of an RCT | To describe the study design and rationale for a patient-centred communication and decision-making intervention for physicians, patients with advanced cancer and their caregivers |

CPR, cardiopulmonary resuscitation; DNACPR, do not attempt cardiopulmonary resuscitation; TEP, treatment escalation plan; NHS, National Health Service; RCT, randomised controlled trial.

**Table 3** Distribution of themes by normalisation process theory (NPT) construct

| NPT core construct [16] | NPT subconstruct and questions [16] | Themes | | | | | |
|---|---|---|---|---|---|---|---|
| | | Input into development | Key clinical proponents | Training and education | Workability/ Functionality | Setting and context | Perceived value and appraisal |
| Coherence | **Differentiation**<br>– Does the intervention differ, and if so how is it different from current practices?<br>– What work has been undertaken to aid understanding of how the new intervention differs from current practice? | | | | | | |
| | **Communal specification**<br>– Do staff have a shared understanding of the purpose of the intervention?<br>– How has a shared understanding of the purpose of the intervention been built among staff within the organisation? | | | | | | |
| | **Individual specification**<br>– Do staff understand their specific tasks and responsibilities in relation to the intervention? What work has been done to ensure they are understood?<br>– Do individuals understand how the intervention will impact on the nature of their work? | | | | | | |
| | **Internalisation**<br>– Do staff understand the potential value of the intervention for their work?<br>– What work has been done to promote understanding of the value, benefits and importance of the new intervention among staff and the organisation as a whole? | | | | | | |
| Cognitive participation | **Initiation**<br>– Are there key individuals identified as those driving the intervention forward? | | | | | | |
| | **Enrolment**<br>– What work has been done to promote others to 'buy-in' and engage with the intervention, including training and education?<br>– Are people open to working with others in new ways to adopt the intervention? | | | | | | |
| | **Legitimation**<br>– Do staff believe that participating in the intervention is a legitimate part of their role?<br>– What work has been undertaken to ensure people believe it is right for them to be involved with the intervention? | | | | | | |
| | **Activation**<br>– Do people continue to support the intervention?<br>– What work has been undertaken to sustain the new intervention in practice and keep people supporting its use? | | | | | | |
| Collective action | **Interactional workability**<br>– Can the intervention be easily integrated into existing work practices?<br>– What work has been undertaken to allow the new intervention to integrate with existing staff and organisational practices? | | | | | | |
| | **Relational integration**<br>– Do individuals feel confident in their own and others' abilities to use the intervention?<br>– What work has been undertaken to build knowledge and confidence in using the new intervention?<br>– Does the intervention impact on working relationships and what work has been done to minimise this? | | | | | | |
| | **Skill set workability**<br>– Has the work (associated with the intervention) been assigned to those with the most appropriate skills?<br>– What formed the decision behind how the division of labour was allocated?<br>– Has sufficient training given staff the skills to enable them to use the intervention? | | | | | | |
| | **Contextual integration**<br>– What resources have been provided to support the new intervention in practice?<br>– Is the intervention adequately supported at managerial level? | | | | | | |

Continued

| NPT core construct[16] | NPT subconstruct and questions[16] | Themes | | | | | |
|---|---|---|---|---|---|---|---|
| | | Input into development | Key clinical proponents | Training and education | Workability/ Functionality | Setting and context | Perceived value and appraisal |
| Reflexive monitoring | **Systemisation**<br>– Has information been collected or are there plans to collect information to determine the usefulness of the intervention through feedback, audit or other means?<br>– Have the effects of the intervention been reported back to those involved?<br>**Communal appraisal**<br>– Is there communal agreement among staff as to the value of the intervention?<br>– Has the worth of the intervention been evaluated collaboratively in formal or informal groups?<br>**Individual appraisal**<br>– What is the effect of the intervention on an individual's workload?<br>– Do individuals value the effect it has on their individual work?<br>**Reconfiguration**<br>– Has appraisal work led to attempts to improve or modify the intervention? | | | | | | |

**Table 3** Continued

perceived the increased training requirement,[31] inclusion within established working practices[39 42 47 51] and feasibility of incorporation into workload[35 42] were important considerations. Repeated sessions were used to train rotational/part-time staff, address staff turnover and enable sustained engagement.[34 36 39 42] Tailored education based on evidence and feedback facilitated the most efficacious training, especially since misunderstanding of responsibilities could result in ineffective or reduced use.[31 32 42 46 53] Training in the application of skills to the intervention was fundamental,[42 47 49] and without this, regardless of skill level, the completion, interpretation and application of interventions in practice could be inconsistent.[44 46 54] Addressing communication skills was highlighted as important in a number of papers.[35 36 38 48 53] Individuals responsible for training could ensure that as far as possible all staff received necessary training.[42 47 49] The provision of guides, algorithms and interactive materials facilitated skill development,[35 47] and patient and carer information materials helped to sustain practice around newly implemented interventions.[31 34 42 45 47 49–52 55–58]

### Intervention workability and functionality

A frequently adopted approach included using existing programme or frameworks (local or otherwise) as the basis for new intervention development.[32 33 37 38 42 54 59] Literature suggested healthcare practices can flex to incorporate interventions designed as part of, or alongside, existing processes.[3 33 36 43 47 49 53] This approach was often more acceptable to staff, promoting incorporation into practice, adherence and behavioural change.[3 33 43 53] Conversely, lack of transferability across healthcare settings was viewed as a barrier to implementation.[37 54] The workability of interventions was important, with the integral use of guidelines and prompts facilitating intervention utility and accessibility.[1 3 31 34–37 39–42 45 47 49 51 53 54 60–65] The use of stickers and brightly coloured forms for insertion into patient medical records promoted easy recognition of interventions.[36 54] For many, successful incorporation into working practices appeared to rely on the paper format of tools, their design and usability.[1 31 45 49 61 62]

### Setting and context

Implementation often occurred within limited clinical settings, and the direct relevance of the specialism (to the intervention) was highlighted as important for promoting its value.[33 37 38 42 60 62–64] Subsequent diffusion from implementation setting to hospital-wide acceptance was demonstrated in relation to a number of interventions.[1 31 36 42] Despite this, multiple barriers and enablers to change were identified, including transferability across primary and secondary care settings,[49 54] staff turnover and management stability,[32] availability of staff for updates, countersignatories to meet completion time frames,[37 53 54] time for communication processes,[33] robustness of implementation sites,[32 42] and the difficulty in transforming clinician attitudes.[38 40 42]

The clinical complexity and unpredictability surrounding the patients for whom the interventions were targeted often affected successful or opportune application.[33 60] Utility was improved where deterioration was predictable,[33 36 60] yet recognition of deterioration was on occasion inherently difficult.[60] Clinical uncertainty meant that at times interventions were not fully utilised for the patients they were designed to support.[33 36]

### Perceived value and appraisal

Staff perceptions and the degree of alignment with the intended purpose of interventions were infrequently reported.[31 40 42] Staff valuing the intervention from the outset had important implications for effective application,[31 33 51 53] and perceived utility, existing supportive local policy and inclusive collaborative development were all highlighted as facilitators.[31–33 35 40–43 45 53] At an organisational level, responsiveness to national guidance and policy was likely to promote value.[3 32 45] Understanding objectives early on in the implementation pathway was key, as misunderstandings were not uncommon[36 44 53] and could lead to reduced or incomplete application of interventions.[36 53]

Following implementation, appraisal of intervention value was frequent,[3 32–34 39 42 54 62–64 66] often demonstrating communal agreement centred around positive impact on working practices.[3 32–34 37 39 40 42 48 54 62 63 66] Increased workload was expressed as a concern by staff, mainly relating to the completion of tools and the need for patient or family discussions.[1 3 37 42 62 63] On appraisal, interventions were perceived to be a worthwhile investment due to patient benefit and the improved clarity and time saved, as a result of decisions and discussions taking place earlier in the care trajectory.[1 3 63] A number of papers recommended or proposed outcome measures relating to patient and relative experience,[3 34–36 39 40 42 54] but only a small number reported utilising such measures.[1 48 56 61]

### DISCUSSION

In this review we have identified a broad and diverse literature focusing on the implementation of goals of care interventions. Findings from this review confirm these interventions are both complex and contentious in nature, and as such it is conceivable that what we have learnt here applies to other contentious interventions and processes, for example discharge planning.

Our analysis has led us to characterise the elements that constitute contentious interventions. Using the example of goals of care interventions, we propose that these interventions consist of three components that intersect at three different levels in a system of negotiated interactions:

1. negotiated decision-making between clinicians, patients and family members, which is localised and characterised by its meaning for the individual
2. the organisational procedure and collective system of making negotiated decisions

3. the sociolegal constraints of taking account of preferences (including consent and capacity) that define the parameters.

Review findings point to negotiated interaction processes taking place between individuals. This mediates a set of procedures about how decisions should be made: a set of expectations about what procedures should be done, what negotiations are possible, and what these look like within and across organisations. Sociolegal conceptualisation involves how organisations must deal with these preferences. All of these continuous components interact and affect a patient's care trajectory.

From the six themes that emerged from our analysis, we have generated transferable learning outcomes for the implementation of contentious interventions. These are described below as a series of propositions relevant to goals of care, which we contend may apply across contentious interventions as a whole. These propositions are the following:

### Individuals resist interventions that replicate the work of existing practices

The value of incorporating interventions into working practices has previously been described[8] and lies in minimising disruption. The adoption of existing intervention formats and integral guidelines was popular, and may act to improve successful compliance and integration through increased staff familiarity and confidence. However there is a fine line, as individuals resist interventions that replicate the work of existing practices.[34 42] Interventions have to be easily differentiated from other practices to be valued. There needs to be clarity regarding the practices that are discontinued, preventing unnecessary duplication, and clear identification of the benefits of the new intervention. Evidence from this review suggests that high visibility methods (eg, a sticker or coloured background) are valued for quick identification and time-saving in clinical practice. These are features unique to the paper-based nature of tools. Technological infrastructure facilitating electronic access to one record in all contexts will enhance the likelihood of widespread embedding across organisations.

### Contentious interventions are difficult to integrate in environments where there is clinical unpredictability and uncertainty

Interventions aimed at improving care for patients facing uncertainty can be difficult to integrate due to the very nature of complexity that exists for these patients and their clinicians. Usually the application of skills and techniques introduces order into such situations, with clinicians seeking scripts to work to. However, clinical uncertainty can render these scripts ineffectual, leading to uncertain outcomes. In light of this, studies in this review suggest the importance of the intervention being delivered by a clinician who has an established relationship with the patient and/or knows their situation well.[37 51] While this represents the ideal, in the reality of clinical practice and

a highly pressurised health service, this may prove difficult to achieve.

## Legitimacy is established where individuals build a shared understanding of purpose that enables them to attribute value to the intervention

The use of senior clinicians and managers in the development process is recognised for building a shared vision.[67] However, the potential for authoritative dominance of certain clinicians is recognised,[68] and efforts must be made to enable active contributions from key members of junior staff. A high degree of clinical ownership is recognised as important for successful implementation.[69] This is more likely to be achieved if all clinicians, irrespective of seniority, believe they have a role to play and are making a valid contribution.[70] Key clinical proponents are essential in building legitimacy, successful and sustained implementation, especially in the context of organisational instability.[71 72]

## Training and education provide the framework for individuals to understand the value of what they do

Training and education have an important role in facilitating a shared understanding of purpose, addressing both moral and technical aspects, which are important for building value in the intervention. Training and education serve to facilitate this, fostering clarity of value in the intervention.

The importance of training and education is not a new concept. However, there is a lack of agreement regarding optimal delivery methods, which may relate to a need for training to fit within the clinical context. Findings from the review point to the need for multimodal educational programmes, repetition and timeliness of training, but this should be considered in the context of staff and financial resource implications.[73 74]

## Appraisal work is critically important because it is how individuals value the intervention

The importance of sharing appraisal outcomes, conveying purpose and value to others is evident, as value is often not realised until successful implementation is achieved. However, reporting of this in the literature was limited.[34 46] In particular, evaluation of patient and family perspectives was notably absent in the papers included in this review. We suggest their inclusion is crucial for imputing value into interventions that seek to include them.[75] Surveys suggest that clinicians will foresee advantages and disadvantages in any proposed intervention.[31 33] Therefore, sharing appraisal work, which addresses the positives and negatives, may be beneficial in providing a more balanced view of an intervention's value.

## The transfer of contentious interventions to other settings is problematic

The initial setting can influence implementation success and subsequent transfer to other locations. In our review there is limited evidence of transfer at scale, over multiple geographical sites.[47] There are multiple barriers through which interventions need to filter, including specialisms, care locations, and structural and cultural factors. In addition, groups of clinicians hold different values, skills and knowledge. This introduces problems of discretion, interpretation and enactment within different contexts and may limit scale-up and transferability. While not widely addressed by the papers in this review, wider implementation literature suggests the importance of involving patients and caregivers when transferring and implementing interventions beyond formal clinical settings.[76 77]

In summary, improving decision-making around goals of care and ensuring patient preferences are taken into account require a better understanding of the implementation processes and factors that promote or impede implementation. We have used the exemplar of goals of care interventions to elucidate propositions in relation to the successful implementation of contentious interventions, where success refers to the routine incorporation and embedding of intervention components into everyday practice. We believe these propositions to be transferable and generalisable beyond the remit of goals of care, to other contentious (and complex) healthcare interventions.

## CONCLUSION

Findings from the review show that while such interventions are variable in design and use, there are a series of collective factors that influence successful implementation into routine clinical practice. We recommend that those seeking to introduce goals of care and other contentious interventions consider the different facets of NPT and use knowledge of this to develop implementation strategies. The contentious nature of these interventions means that their incorporation into everyday practice is dependent on a number of factors. Building a shared understanding of purpose that enables participants to attribute value to the intervention is key, and both training and education and appraisal work, play an important role in this process. Identifying clinical proponents who are able to, not only drive, but positively influence implementation is essential. Implementing complex and contentious interventions presents challenges that operate at an individual, organisational and systems level. It is these interaction level processes that make an intervention contentious, as well as making it challenging to introduce and embed. Success is more likely to occur, be established and sustained, if due attention is paid to the processes that facilitate operationalisation in the healthcare settings in which implementation occurs.

**Acknowledgements** The authors would like to thank Dr Mark Banting for his assistance with data extraction in the initial phase of the review.

**Contributors** MM, CRM and AR designed the review. AC performed searches. AC and CM screened titles and abstracts, and full paper screening was carried out by AC, MM and CM. Data extraction was performed by AC, MM and SL. AC and MM drafted the manuscript with assistance from SL, CM, NC and AR. All authors critically reviewed the manuscript for intellectual content and approved the final version of the paper. AC is the guarantor.

**Funding** This work was supported by the National Institute for Health Research Collaboration for Leadership in Applied Health Research and Care (NIHR CLAHRC) Wessex which is a partnership between Wessex NHS organisations and partners and the University of Southampton. Funders had no role in study design, data collection and analysis, decision to publish or preparation of the manuscript

**Disclaimer** The views expressed are those of the author(s) and not necessarily those of the NHS, the NIHR or the Department of Health.

**Competing interests** None declared.

**Provenance and peer review** Not commissioned; externally peer reviewed.

**Data sharing statement** No additional data are available.

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
