## [Reviewer comments · BMJ Open]

ARTICLE DETAILS

TITLE (PROVISIONAL)	Implementing communication and decision-making interventions directed at goals of care: a theory led scoping review
AUTHORS	Cummings, Amanda; Lund, Susi; Campling, Natasha; May, Carl; Richardson, Alison; Myall, Michelle

VERSION 1 – REVIEW

REVIEWER	Carole Mockford Warwick Medical School University of Warwick
REVIEW RETURNED	05-May-2017

GENERAL COMMENTS	An interesting and well thought through scoping review which will be of interest to those conducting implementation studies. One point that is always an issue with the time it takes to conduct a review of this size is keeping it updated. This is a growing area of interest so it would be interesting to acknowledge any key advancements, if there are any, since your 2015 cut off for literature.
--

REVIEWER	Lisa Jane Brighton Cicely Saunders Institute, King's College London United Kingdom
REVIEW RETURNED	30-Jun-2017

GENERAL COMMENTS	This paper reports a scoping review aiming to identify factors that are important in the implementation of goals of care tools/interventions. The authors have clearly conducted a thorough, theoretically driven piece of work. I believe their key themes summarised in the results will be helpful for those designing, implementing, and testing complex interventions in healthcare settings. However, there are some key areas of the paper that require strengthening. Introduction: (a) The introduction is long and may benefit from cutting and re-ordering (e.g. moving the aim to the end of the introduction) Methods: (b) The inclusion and exclusion criteria require more clarification, as it is currently difficult to understand what would and would not be included. For example, what comprises a 'communication
---

guideline'? Some introduction text could be moved to this section. It may help to use something similar to the PCC structure: Population; Concept; Context. Could the authors also please clarify why interventions focused on DNACPR only were not included?

(c) Given the need to search a diverse and broad body of evidence, could the authors please clarify why only two scientific databases were searched? Did the authors use citation tracking or search the reference lists of included papers?

(d) I would like to see an example of a full search of one database (e.g. Medline) as entered line by line, to allow replication with word variations / wildcard functions etc. Please could the authors also clarify why they used 'clinical care outreach team' for a set of their searches, with very minimal results, rather than simply 'critical care'?

(e) Please can the authors provide more detail on the data included in their content analysis. In all cases, is this an analysis of the original paper authors' own reflections, or does it sometimes include qualitative data (e.g. from interviews with staff and patients around implementation)? If there is a mixture, it would be helpful for the authors to include in Table 2 what types of data were contributed by each study.

(f) The analysis method also requires clarification. I understand the content analysis using the NPT constructs, but please could the authors explain how the final six themes in the results were then generated?

Results:

(g) In Table 2, please could the authors provide a little more information about the interventions, and context beyond just setting (e.g. are they with a specific ward type, staff group, or patient group)

(h) Whilst I can see how the NPT model was helpful in extracting a complete set of data on each intervention, I do not think it helps with understanding the key themes. For a new reader it is very hard to navigate the constructs, sub-constructs, findings and themes in Table 3, and then how this maps to the results text. I believe the results would be much stronger if they were to only focus on the themes identified in response to the authors' review aim; these are well explained and of practical use to people working with these interventions. If the authors are keen to display the overlap with NPT constructs, I would recommend swapping the table in Appendix 4 instead of Table 3.

Discussion:

(i) I do not think the section on elements that constitute a contentious intervention are relevant to the authors' review aims, and should therefore be removed. Throughout this paper, referring to these interventions as 'complex' would suffice.

(j) At present, it is difficult for the reader to match the authors' 6 themes from their results to the 6 recommendations they supply. Matching the ordering of these and subtitle language between the results and discussion could help this.

VERSION 1 – AUTHOR RESPONSE

Reviewer 1 Comments:

An interesting and well thought through scoping review which will be of interest to those conducting implementation studies. One point that is always an issue with the time it takes to conduct a review of this size is keeping it updated. This is a growing area of interest so it would be interesting to acknowledge any key advancements, if there are any, since your 2015 cut off for literature.

Thank you for highlighting a key point. To justify our August 2015 cut-off and highlight major developments since then, we have added information regarding the national RESPECT work (and how our review can help to inform this) in the Methods section.

Reviewer 2 Comments:

This paper reports a scoping review aiming to identify factors that are important in the implementation of goals of care tools/interventions. The authors have clearly conducted a thorough, theoretically driven piece of work. I believe their key themes summarised in the results will be helpful for those designing, implementing, and testing complex interventions in healthcare settings. However, there are some key areas of the paper that require strengthening.

Introduction:

(a) The introduction is long and may benefit from cutting and re-ordering (e.g. moving the aim to the end of the introduction)

Thank you for your suggestion. Whilst we believe the length of the introduction is appropriate in providing adequate background information, we agree that the aim is better placed at the end of the introduction, and have moved this paragraph accordingly. We have also amended the aim to reflect our incorporation of goals of care as contentious interventions, which we believe, makes the purpose of the paper clearer (in response to comment (i)).

Methods:

(b)(i) The inclusion and exclusion criteria require more clarification, as it is currently difficult to understand what would and would not be included. For example, what comprises a 'communication guideline'?

We were uncertain as to exactly what the reviewer is referring to, as 'communication guideline' is not within the inclusion and exclusion criteria. If this relates to Table 2, then communication guidelines around goals of care (not specific to one particular decision of interest) were identified from the literature. We have amended the description to 'communication guidelines around goals of care' to improve clarity.

(ii) Some introduction text could be moved to this section. It may help to use something similar to the PCC structure: Population; Concept; Context.

Please see our response to comment (g).

(iii) Could the authors also please clarify why interventions focused on DNACPR only were not included?

Please see paragraph two of the Introduction, which describes how goals of care interventions are often an extension of DNACPR process. In addition, we have added text to clarify the reasons for non-inclusion of DNACPR only papers within the Inclusion and Exclusion Criteria.

(c) Given the need to search a diverse and broad body of evidence, could the authors please clarify why only two scientific databases were searched? Did the authors use citation tracking or search the reference lists of included papers?

We have undertaken a scoping review and not a systematic review. On this basis we would argue that extensive searches within the two main (most relevant) databases, as well as grey literature searches, are sufficient.

(d) I would like to see an example of a full search of one database (e.g. Medline) as entered line by line, to allow replication with word variations / wildcard functions etc. Please could the authors also clarify why they used 'clinical care outreach team' for a set of their searches, with very minimal results, rather than simply 'critical care'?

Please see Appendix 1 for a full outline of our primary literature search strategy, which outlines all combinations of our key word searches. This review was never intended to be a systematic review, and as such our searches are consistent with a scoping review methodology. As described within the Methods section, key words were explored and refined during a two-stage search process. The key word search for 'critical care outreach team' was used specifically due to the role of such clinical teams who are called upon to make decisions relating to treatment escalation.

(e) Please can the authors provide more detail on the data included in their content analysis. In all cases, is this an analysis of the original paper authors' own reflections, or does it sometimes include qualitative data (e.g. from interviews with staff and patients around implementation)? If there is a mixture, it would be helpful for the authors to include in Table 2 what types of data were contributed by each study.

We have amended Table 2 to include more detail on the study design and methods. This now outlines the type of data contributed by each study and therefore included within our content analysis.

(f) The analysis method also requires clarification. I understand the content analysis using the NPT constructs, but please could the authors explain how the final six themes in the results were then generated?

Thank you for this helpful comment. We have made some amendments to the Data Analysis text to clarify our analytical processes. Each stage of our analysis process is now outlined more clearly.

Results:

(g) In Table 2, please could the authors provide a little more information about the interventions, and context beyond just setting (e.g. are they with a specific ward type, staff group, or patient group)

We have added more specific detail to the healthcare settings for each paper (where available) in Table 2.

(h) Whilst I can see how the NPT model was helpful in extracting a complete set of data on each intervention, I do not think it helps with understanding the key themes. For a new reader it is very hard to navigate the constructs, sub-constructs, findings and themes in Table 3, and then how this maps to the results text. I believe the results would be much stronger if they were to only focus on the themes identified in response to the authors' review aim; these are well explained and of practical use to people working with these interventions. If the authors are keen to display the overlap with NPT constructs, I would recommend swapping the table in Appendix 4 instead of Table 3.

Thank you for your suggestion. After reviewing the paper in response to your comment, we have moved Table 3 to the Appendix (now Appendix 4), and have adapted the table in Appendix 4 to incorporate the questions relating to each NPT sub-construct (for reader clarity) and included this in place of the original Table 3. We believe this has improved the readability of the paper and drawn the focus of the Results section to the identified 6 themes, which as you state, have a more practical application for our readership.

Discussion:

(i) I do not think the section on elements that constitute a contentious intervention are relevant to the authors' review aims, and should therefore be removed. Throughout this paper, referring to these interventions as 'complex' would suffice.

Thank you for your suggestion. However, a key outcome of this paper is the identification and characterisation of goals of care interventions as contentious as well as complex. This is argued throughout the paper and we have defined the difference between complex and contentious interventions. In response to your comment we have amended our aim to include this key outcome.

(j) At present, it is difficult for the reader to match the authors' 6 themes from their results to the 6 recommendations they supply. Matching the ordering of these and subtitle language between the results and discussion could help this.

Thank you for this helpful comment. In response, we have added text to our Data Analysis section to clarify the analytical stages that led to the development of our propositions relevant to the implementation of contentious interventions. The propositions represent a higher level of analysis, generated from the themes outlined in our Results. Whilst they are generated from them, they are not supposed to match the 6 themes. To help provide clarity, we have removed the numbering of the propositions in the Discussion, and have amended the introductory text that precedes these propositions.

VERSION 2 – REVIEW

REVIEWER	Lisa Jane Brighton Cicely Saunders Institute, King's College London
REVIEW RETURNED	09-Aug-2017

GENERAL COMMENTS	Thank you to the authors for their clear and detailed response to the reviewer comments. The paper has improved as a result of their changes, and where changes were not made the justification is clear.
---

	Only one very minor comment that could be addressed: at the very start of the discussion the authors say 'findings from this review suggest these interventions are both complex and contentious in nature', however this was decided at the outset of the review. This could simply be amended to say 'findings from this review may apply to other contentions interventions and processes' etc.
--	--

VERSION 2 – AUTHOR RESPONSE

Reviewer 2 Comment:

Only one very minor comment that could be addressed: at the very start of the discussion the authors say 'findings from this review suggest these interventions are both complex and contentious in nature', however this was decided at the outset of the review. This could simply be amended to say 'findings from this review may apply to other contentions interventions and processes' etc.

Thank you for drawing this to our attention. We agree that as we have already introduced goals of care interventions as complex and contentious in our introduction, this sentence should be re-phrased. We have changed it accordingly.